# Enhanced Proliferative and Osteogenic Potential of Periodontal Ligament Stromal Cells

**DOI:** 10.3390/biomedicines11051352

**Published:** 2023-05-03

**Authors:** Laura Alves, Vanessa Machado, João Botelho, José João Mendes, Joaquim M. S. Cabral, Cláudia L. da Silva, Marta S. Carvalho

**Affiliations:** 1Department of Bioengineering and iBB—Institute for Bioengineering and Biosciences, Instituto Superior Técnico, Universidade de Lisboa, Av. Rovisco Pais, 1049-001 Lisboa, Portugal; 2Associate Laboratory i4HB—Institute for Health and Bioeconomy, Instituto Superior Técnico, Universidade de Lisboa, Av. Rovisco Pais, 1049-001 Lisboa, Portugal; 3Clinical Research Unit, Egas Moniz Center for Interdisciplinary Research, Egas Moniz School of Health and Science, 2829-511 Almada, Portugal; 4Evidence-Based Hub, Egas Moniz Center for Interdisciplinary Research, Egas Moniz School of Health and Science, 2829-511 Almada, Portugal

**Keywords:** adipose tissue, bone marrow, mesenchymal stromal cells, osteogenic potential, periodontal ligament stromal cells, periodontitis, periodontium

## Abstract

Cell-based therapies using periodontal ligament stromal cells (PDLSC) for periodontal regeneration may represent an alternative source for mesenchymal stromal cells (MSC) to MSC derived from bone marrow (MSC(M)) and adipose tissue (MSC(AT)). We aimed to characterize the osteogenic/periodontal potential of PDLSC in comparison to MSC(M) and MSC(AT). PDLSC were obtained from surgically extracted healthy human third molars, while MSC(M) and MSC(AT) were obtained from a previously established cell bank. Flow cytometry, immunocytochemistry, and cell proliferation analyses provided cellular characteristics from each group. Cells from the three groups presented MSC-like morphology, MSC-related marker expression, and multilineage differentiation capacity (adipogenic, chondrogenic, and osteogenic). In this study, PDLSC expressed osteopontin, osteocalcin, and asporin, while MSC(M) and MSC(AT) did not. Of note, only PDLSC expressed CD146, a marker previously applied to identify PDLSC, and presented higher proliferative potential compared to MSC(M) and MSC(AT). Upon osteogenic induction, PDLSC exhibited higher calcium content and enhanced upregulation of osteogenic/periodontal genes compared to MSC(M) and MSC(AT), such as *Runx2*, *Col1A1* and *CEMP-1*. However, the alkaline phosphatase activity of PDLSC did not increase. Our findings suggest that PDLSC might be a promising cell source for periodontal regeneration, presenting enhanced proliferative and osteogenic potential compared to MSC(M) and MSC(AT).

## 1. Introduction

Conditions affecting the periodontium are among the most prevalent worldwide and have been increasing in the last three decades [1,2]. The most relevant is periodontitis, an inflammatory non-communicable disease characterized by the progressive destruction of tooth-supporting tissues [2]. Triggered by a dysbiotic dental plaque accumulation, periodontitis affects the quality of life, with aesthetic implications, occasional pain, discomfort, and impaired mastication [3,4]. Beyond that, periodontitis also comes with socioeconomic impacts [5]. If untreated, periodontitis can lead to bone destruction, damaging both hard and soft periodontal tissues, resulting in tooth attachment and bone loss [6]. Additionally, its impact on systemic health is consistently proven [7], thus making its treatment and resolution of its sequelae a priority.

Regenerative periodontal surgical treatments intend to re-establish lost periodontal tissues. In efforts to guide and increase the attachment of the teeth to the periodontium and induce bone gain, several surgical approaches have been proposed. Recent advances rely on guided tissue regeneration strategies using resorbable membranes, grafts and/or scaffolds [8]. Furthermore, scaffold-based therapies allow the delivery of cells and proteins directly to the defect site. However, despite these efforts, regeneration of a functional periodontium has never been achieved [8]. Recently, the potential use of cell therapies, in particular mesenchymal stromal cells (MSC), for periodontal tissue engineering applications has been explored [9,10]. Thus, understanding the source and the distinct differentiation patterns of these cells is crucial to elucidate their potential therapeutic use.

MSC have appeared as an appealing progenitor cell source for bone tissue engineering applications [11]. Originally isolated from bone marrow, MSC can also be derived from several adult tissues, including adipose tissue, umbilical cord, and Wharton’s jelly [12]. However, most of these sources require invasive procedures to isolate MSC with limited availability. As an alternative, dental MSC are an interesting source with less associated invasiveness [13]. Among dental MSC, periodontal ligament stromal cells (PDLSC) can potentially act as a source of renewable progenitor cells, capable of producing cementoblasts, osteoblasts and MSC [14]. Seo and colleagues successfully isolated PDLSC with stem/stromal cell characteristics from the human periodontal ligament (PDL) [13]. PDLSC have been previously compared to bone marrow-derived MSC (MSC(M)) as they are widely used in clinical studies [15]. Compared to MSC(M), PDLSC show a similar phenotypic profile and the capacity to develop into cementoblast-like cells, osteoblasts and adipocytes in vitro [13]. In vivo results also demonstrated that, upon transplantation, PDLSC formed cementum/PDL-like tissues in immunocompromised mice [13]. Moreover, PDLSC have demonstrated immunomodulatory and anti-inflammatory potential [15]. However, challenges remain to be addressed concerning the identification and characterization of PDLSC, in particular, aiming to find specific markers associated with PDL. Human PDLSC were first isolated from PDL heterogeneous cell populations using two early MSC-related cell surface markers: Stro-1 and CD146 [13]. Although Stro-1^+^ MSC were able to form clonogenic fibroblast colonies in vitro [16], the use of Stro-1 alone for cell isolation is not sufficient to obtain pure populations of PDLSC. In fact, the reported percentage of Stro-1^+^ cells in the PDL tissue varies widely between 1.2% and 33.5% [17,18].

Since its isolation, studies on PDLSC reported the expression of several additional cell surface markers. However, more data regarding PDLSC characterization is required to confirm PDLSC identity and assess their use in clinical studies and stem cell therapy in dentistry. On the other hand, the osteogenic potential of PDLSC remains to be investigated: in particular, their performance compared to MSC derived from other adult tissues, such as adipose tissue and bone marrow. Therefore, the present study aims to explore the proliferative and osteogenic capacity of PDLSC for applications in periodontal regeneration. For this, we characterized and compared the osteogenic/periodontal potential of MSC derived from different adult sources, such as bone marrow, adipose tissue, and PDL. In this study, we compared the morphology, immunophenotype, proliferation rate, and trilineage differentiation potential of adipose tissue-derived MSC (MSC(AT)), MSC(M), and PDLSC. Moreover, the osteogenic potential and gene expression analysis of these cells were compared to evaluate their clinical application potential in repairing bone defects. We believe that this work may provide important data to support the selection of the most promising cells for periodontal tissue engineering applications.

## 2. Materials and Methods

### 2.1. Isolation and Culture of Human PDLSC

Healthy human third molars were extracted for orthodontic reasons from two healthy male patients (20 and 28 years old) by dentists from the Egas Moniz Dental Clinic, a university dental clinic located at Egas Moniz School of Health and Science (Almada, Portugal). Samples were washed with phosphate-buffered saline solution (PBS, Gibco, Thermo Fisher Scientific, Bohemia, NY, USA) supplemented with 1% antibiotic-antimycotic (A/A, Gibco, Thermo Fisher Scientific) (washing buffer). Then, the PDL was gently separated from the surface of the root and plated into a six-well plate containing a washing buffer. The solution containing the PDL tissue was transferred into a 15 mL tube and centrifuged at 1500 rpm for 10 min. After discarding the supernatant, cells were digested in a solution of 3 mg/mL collagenase type I (Sigma-Aldrich, St. Louis, MO, USA) and 4 mg/mL dispase (Sigma-Aldrich) for 1 h at 37 °C. To neutralize enzyme activity, an excess volume of washing buffer was added to the digested tissue and was then strained through a 70 μm cell strainer to remove undigested tissue from cells. Cells were centrifuged at 1500 rpm for 10 min and resuspended in 2 mL of low-glucose Dulbecco’s modified Eagle’s medium (DMEM, Gibco, Thermo Fisher Scientific) supplemented with 10% fetal bovine serum (FBS MSC qualified, Gibco, Thermo Fisher Scientific) and 1% A/A. Upon reaching confluency, adherent PDLSC were harvested using a solution of 0.05% trypsin (Gibco, Thermo Fisher Scientific) and counted using the Trypan Blue exclusion method (Gibco, Thermo Fisher Scientific). Cells were then plated on T-75 flasks using low-glucose DMEM supplemented with 10% FBS and 1% A/A and kept at 37 °C, 5% CO_2_ in a humidified atmosphere. The medium renewal was performed every three to four days.

### 2.2. Culture of MSC(M) and MSC(AT)

MSC(M) and MSC(AT) were isolated and expanded from healthy donors (male, 32–46 years old) according to previously established protocols [19,20]. Human tissue samples were obtained from hospitals under a collaboration agreement with the Institute for Bioengineering and Biosciences at Instituto Superior Técnico (bone marrow: Instituto Português de Oncologia Francisco Gentil, Lisboa; adipose tissue: Clínica de Todos-os-Santos, Lisboa). Samples were obtained from healthy donors after written informed consent according to Directive 2004/23/EC of the European Parliament and of the Council of 31 March 2004 on setting standards of quality and safety for the donation, procurement, testing, processing, preservation, storage and distribution of human tissues and cells (Portuguese Law 22/2007, 29 June), with the approval of the Ethics Committee of the respective clinical institution. MSC(M) and MSC(AT) were plated using low-glucose DMEM supplemented with 10% FBS and 1% A/A. The medium was changed every three to four days. Two donors from each cell type (PDLSC, MSC(M) and MSC(AT)) were used in this work. The study design is shown in Figure 1.

### 2.3. Multilineage Differentiation and Stainings

To investigate the multipotency of MSC derived from different sources, in vitro differentiation studies (osteogenic, adipogenic, and chondrogenic lineages) were performed. Cells that were not cultured under differentiation conditions (i.e., expansion medium, DMEM + 10% FBS) were used as negative controls.

#### 2.3.1. Osteogenic Differentiation

PDLSC, MSC(M), and MSC(AT) were cultured at 3000 cells/cm^2^ on 24-well plates with DMEM + 10% FBS + 1% A/A. At 80% confluence, cells were incubated with osteogenic differentiation medium composed of low-glucose DMEM supplemented with 10% FBS and 1% A/A, 10 mM β-glycerophosphate (Sigma-Aldrich), 10 nM dexamethasone (Sigma-Aldrich), and 50 μg/mL ascorbic acid-2-phosphate (Sigma-Aldrich). After 21 days of osteogenic stimuli, cells were fixed in a 4% paraformaldehyde (PFA) (Sigma-Aldrich) solution and stained for alkaline phosphatase (ALP), von Kossa (VK), and Alizarin Red stainings as previously described [11].

##### Alkaline Phosphatase (ALP) Activity Assay

After 21 days of osteogenic differentiation, ALP activity was evaluated using a colorimetric ALP kit (BioAssays Systems, Hayward, CA, USA) according to the manufacturer’s protocol. Firstly, samples were washed with PBS and incubated in lysis buffer (0.1% Triton X-100 in PBS) by shaking for 30 min at room temperature. Then, the lysate was added to a p-nitrophenyl phosphate solution (10 mM). Absorbance at 405 nm was measured with a plate reader and normalized to the metabolic activity. Three different samples were used for each condition, and absorbance was measured in triplicate. The metabolic activity of MSC was evaluated using AlamarBlue^®^ cell viability reagent (Thermo Fisher Scientific) following the manufacturer’s guidelines. A 10% AlamarBlue^®^ solution was added to cells and incubated for 3 h at 37 °C. Fluorescence intensity was measured in a plate reader (Infinite M200 Pro, Tecan, Männedorf, Switzerland) at an excitation/emission wavelength of 560/590 nm. Three samples were used for each condition, and fluorescence measurements were performed in triplicate.

##### Calcium Quantification Assay

After 21 days of osteogenic differentiation, samples were incubated with a 0.5 M HCl solution (Sigma-Aldrich) by shaking overnight at 4 °C. Total calcium content was determined using a calcium colorimetric assay kit (Sigma-Aldrich), according to manufacturer’s instructions. Calcium standard solutions were prepared in parallel. Absorbance at 575 nm was measured in triplicate for each condition and normalized to the metabolic activity. Three samples were used for each condition.

#### 2.3.2. Adipogenic Differentiation

For adipogenic differentiation, PDLSC, MSC(M), and MSC(AT) were cultured at 3000 cells/cm^2^ in expansion conditions until 80% confluency. After 21 days under adipogenic differentiation medium (StemPro™ Adipogenesis Differentiation Kit, Gibco, Thermo Fisher Scientific), cells were fixed with 4% PFA and stained with Oil Red O solution, as previously described [11].

#### 2.3.3. Chondrogenic Differentiation

For chondrogenic differentiation, cells were cultured as cell aggregates on ultra-low attachment plates (Corning), as previously reported [11]. Cells were maintained in a chondrogenic differentiation medium (MesenCult™ Chondrogenic Differentiation Kit, Stemcell Technologies, Vancouver, BC, Canada) for 21 days, and chondrogenic differentiation was confirmed with Alcian Blue staining (Sigma-Aldrich) [11].

### 2.4. Flow Cytometry Analysis

Immunophenotypic analysis of PDLSC, MSC(M) and MSC(AT) was performed by flow cytometry to assess the expression of CD14, CD19, CD29, CD34, CD44, CD45, CD73, CD80, CD90, CD105, CD106, CD146, CD166, HLA-DR, and STRO-1 (BioLegend, San Diego, CA, USA). Cells were incubated with each mouse anti-human monoclonal antibody, as previously reported [11]. A minimum of 10,000 events were collected, and flow cytometric analysis was performed using FACScalibur flow cytometer (Becton Dickinson, Franklin Lakes, NJ, USA) and CellQuestTM software (Becton Dickinson) was used for acquisition. Lastly, data analysis was conducted using Flowing Software (University of Turku, Turku, Finland).

### 2.5. Immunocytochemistry Analysis

The distribution of several ECM proteins, such as collagen I (Col I), asporin, fibronectin, laminin, osteopontin, osteocalcin, cementum protein 1, and Stro-1, was analyzed in MSC(M), MSC(AT), and PDLSC. After PFA fixation, cells were blocked and permeabilized for 45 min in 1% bovine serum albumin (BSA, Sigma-Aldrich), 10% FBS, and 0.3% Triton X-100 (Sigma-Aldrich). Primary antibodies (dilution 1:500 in 1% BSA, 10% FBS, and 0.3% Triton X-100), including rabbit anti-human collagen I (Thermo Fisher Scientific), asporin (Abcam, Cambridge, UK), laminin (Thermo Fisher Scientific), osteopontin (Abcam), osteocalcin (Sigma-Aldrich), cementum protein 1 (Abcam) and mouse anti-human collagen IV (Thermo Fisher Scientific), fibronectin (Thermo Fisher Scientific) and stro-1 (Thermo Fisher Scientific) were incubated overnight at 4 °C. After washing with 1% BSA, goat anti-mouse IgG Alexa Fluor 546, goat anti-rabbit IgG Alexa Fluor 546 and goat anti-mouse IgG Alexa Fluor 488 (Thermo Fisher Scientific, dilution 1:200 in 1% BSA solution) were used as secondary antibodies and incubated for 1 h at room temperature. Finally, cell nuclei were stained with DAPI (Thermo Fisher Scientific, 1.5 μg/mL) for 5 min.

### 2.6. Cell Morphology Assays

Cells were seeded on 24-well plates at 3000 cells/cm^2^, and cell morphology was assessed at different time points. Cytoskeleton actin filaments were stained with Phalloidin-TRITC (Sigma-Aldrich; dilution 1:250, 2 μg/mL) for 45 min in the dark. Afterward, nuclei were stained with DAPI (1.5 μg/mL) for 5 min.

### 2.7. Proliferation Assays

PDLSC, MSC(M), and MSC(AT) were cultured for 9 days on 12-well plates at different cell densities: 1500 cells/cm^2^ and 3000 cells/cm^2^ in expansion conditions (DMEM + 10% FBS). At each time point, cells were harvested using a 0.05% trypsin solution (Gibco, Thermo Fisher Scientific) and counted using the Trypan Blue exclusion method to determine cell numbers.

### 2.8. Quantitative Reverse Transcription-Polymerase Chain Reaction (qRT-PCR) Analysis

After 21 days of osteogenic differentiation, PDLSC, MSC(M) and MSC(AT) total RNA was extracted using RNeasy Mini Kit (QIAGEN, Hilden, Germany), followed by cDNA synthesis with High-Capacity cDNA Reverse Transcription kit (Life Technologies, Carlsbad, CA, USA). Primer sequences used are summarized in Table 1, and qRT-PCR was performed using NZYSpeedy qPCR Green Master Mix (2×), ROX plus (NZYTech, Lisbon, Portugal) and StepOnePlus real-time PCR system (Applied Biosystems, Waltham, MA, USA). Reactions were performed in triplicate and carried out for 10 min at 95 °C, followed by 40 cycles of 15 sec at 95 °C and 1 min at 60 °C. Gene expression was normalized to the housekeeping gene glyceraldehyde 3-phosphate dehydrogenase (GAPDH), and fold-change was calculated considering baseline expression at day 0 (undifferentiated cells). A threshold cycle (Ct) was observed in the exponential phase. ΔΔCt values were calculated using geometric means and expressed as 2^−ΔΔCt^.

### 2.9. Statistical Analysis

For each experiment, two different donors of each cell source (MSC(M), MSC(AT) and PDLSC) were used, and three independent experiments (cells from different passages, P3–P5) for each donor were performed with three technical replicates of each. The statistical analysis of the data was performed using one-way ANOVA, followed by the Tukey post-hoc test. GraphPad Prism version 7 software was used in the analysis, and data was considered to be significant when the *p*-values obtained were less than 0.05 (95% confidence intervals, * *p* < 0.05).

## 3. Results

### 3.1. Characterization of MSC Derived from Different Adult Tissue Sources: Periodontal Ligament, Bone Marrow and Adipose Tissue

In order to identify the immunophenotypic differences between MSC isolated from different sources, comparative studies with MSC(M), MSC(AT) and PDLSC were performed by flow cytometry for cells cultured for passages 3, 5, and 7 (Figure 1, Appendix A).

All cells displayed strongly positive expression (>90%) of MSC-associated cell surface markers CD29, CD44, CD73, CD90 and CD105 in passage 3. Results showed that CD166 expression was positive for PDLSC (>86.1%) and MSC(M) (>71.3%) in passages 3, 5, and 7 (Figure 1). Additionally, when compared with MSC(AT), PDLSC presented a statistically significant enhancement of CD166 expression in all passages studied. When comparing CD166 expression between MSC(M) and MSC(AT), results showed that MSC(M) presented higher CD166 expression regardless of passage, with statistically significant differences in passages 3 and 5. Thus, CD166 expression of MSC(AT) was lower for all passages in comparison with both PDL- and bone marrow-derived counterparts. Moreover, the CD166 expression levels of MSC(AT) decreased with increasing passages, presenting an expression of (90.45 ± 0.33)% in passage 3, (45.94 ± 2.65)% in passage 5, and (64.87 ± 5.12)% in passage 7. CD106 and Stro-1 expression levels were low (<10%) in all samples analyzed in passages 3, 5, and 7, regardless of the tissue source (Figure 1). As expected, lack of expression of hematopoietic cell-associated markers (CD14, CD19, CD34, CD45, and HLA-DR) was also verified for PDLSC, MSC(M) and MSC(AT) in passages 3, 5, and 7. Additionally, the immune cell-related marker CD80 presented low expression levels by samples from all MSC sources. Lastly, results showed that CD146, a marker previously used to identify PDLSC, was only expressed by PDLSC in passages 3, 5, and 7. Despite this, results showed that CD146 expression by PDLSC decreased with passaging, expressing (67.40 ± 0.91)% in passage 3, (69.51 ± 1.63)% in passage 5, and (24.06 ± 1.23)% in passage 7 (Figure 1).

Despite presenting some differences, cells from the different tissues exhibited similar spindle-shaped morphology (Figure 2). After seven days of culture, PDLSC cultures were completely confluent, contrary to bone marrow and adipose tissue-derived cultures, as observed by DAPI/Phalloidin staining (Figure 2).

To further explore differences in protein expression between PDLSC, MSC(M) and MSC(AT), a comparative immunocytochemistry analysis was performed (Figure 3). Results confirmed the expression of the common extracellular matrix (ECM) proteins, such as laminin and fibronectin, in all samples. Additionally, expression of the known MSC marker Stro-1, asporin, osteocalcin (OC) and osteopontin (OPN) was only detected on undifferentiated PDLSC and not detected in undifferentiated MSC(M) and MSC(AT) (Figure 3). The positive stainings of osteogenic markers OPN and OC suggested that PDLSC might have higher osteogenic potential, even when not cultured under osteogenic differentiation conditions (i.e., expansion medium).

### 3.2. Proliferative Potential of MSC Derived from Periodontal Ligament, Bone Marrow and Adipose Tissue

To assess the proliferative capacity of MSC from different sources (PDLSC, MSC(M) and MSC(AT)), cells (P3–P5) were plated at different cell seeding densities: 1500 cells/cm^2^ and 3000 cells/cm^2^ (Figure 4).

Under the conditions of our study, PDLSC presented a significantly higher proliferative rate in comparison with MSC(M) and MSC(AT). After nine days of culture at 3000 cells/cm^2^, a significant increase in cell numbers was observed for PDLSC compared to MSC(M) and MSC(AT) (Figure 4A). PDLSC reached a cell number of (7.90 ± 1.60) × 10^5^, whereas MSC(M) and MSC(AT) only reached a cell number of (0.54 ± 0.22) × 10^5^ and (0.83 ± 0.28) × 10^5^, respectively. When cells were seeded at 1500 cells/cm^2^, PDLSC reached a cell number of (6.30 ± 0.79) × 10^5^, while MSC(M) and MSC(AT) presented cell numbers of (0.29 ± 0.21) × 10^5^ and (0.94 ± 0.01) × 10^5^, respectively (Figure 4B). Additionally, MSC(AT) presented higher cell numbers compared to MSC(M) (Figure 4).

During the nine-day culture period, all cells exhibited similar cell growth behavior regardless of cell seeding density. Moreover, after nine days in culture, PDLSC reached statistically significant higher population doublings compared to MSC(M) and MSC(AT) (6.0 ± 0.3 vs. 2.1 ± 0.6 vs. 2.8 ± 0.5 for PDLSC, MSC(M), and MSC(AT), respectively, when cultured at 3000 cells/cm^2^) (Figure 4C). PDLSC exhibited higher cell growth rates regardless of seeding density (Appendix A).

### 3.3. Osteogenic Potential of MSC Derived Periodontal Ligament, Bone Marrow and Adipose Tissue

After 21 days under differentiation conditions towards adipogenic, chondrogenic, and osteogenic lineages, stainings confirmed the successful in vitro trilineage differentiation of PDLSC, MSC(M), and MSC(AT) (Figure 5) at passages 3 and 5. Figure 5 revealed positive stainings of adipocytes, osteoblasts, and chondrocytes with Oil red O, Alkaline Phosphatase (ALP)/von Kossa (VK) and Alcian blue stainings, respectively. Cells cultured under expansion conditions (DMEM) were used as negative controls. Although stainings were not quantified, qualitative studies on multilineage differentiation were able to suggest some differences between the three cell sources. Regarding adipogenic differentiation, PDLSC exhibited a lower amount of Oil Red O-stained liquid droplets compared with MSC(M) and MSC(AT), suggesting a decreased adipogenic potential. Interestingly, osteogenic differentiation stainings revealed that PDLSC produced a lower amount of ALP compared with MSC(M) and MSC(AT) (Figure 5). MSC(M) and MSC(AT) multilineage differentiation assays presented similar ALP/VK, Alizarin Red, and Oil Red-O stainings (Figure 5).

To evaluate the impact of tissue source on the in vitro osteogenic potential of MSC, calcium deposition, ALP activity and osteogenic/periodontal gene expression were evaluated after 21 days of culture under osteogenic differentiation conditions (P3–P5). As expected, results showed that cells from all sources cultured under osteogenic induction conditions (OSTEO) presented higher calcium accumulation compared to cells cultured under expansion conditions (DMEM) (Figure 6A). These differences confirmed the successful osteogenic differentiation of MSC from different sources. A statistically significant enhancement in calcium accumulation was observed for MSC derived from PDL compared to MSC(M) and MSC(AT) ([17.17 ± 2.02] × 10^−7^ μg·μL^−1^) (Figure 6A). However, MSC(M)and MSC(AT) did not present statistically significant differences in calcium accumulation after 21 days of osteogenic differentiation (MSC(M): [9.20 ± 1.21] × 10^−7^ μg·μL^−1^, MSC(AT): [11.74 ± 0.35] × 10^−7^ μg·μL^−1^). These results demonstrated a higher mineralization capacity from PDLSC compared to bone marrow and adipose tissue-derived cells under the conditions of our study.

Furthermore, results demonstrated that ALP activity of PDLSC did not increase after osteogenic differentiation ([1.40 ± 0.13] × 10^−4^ μg·μL^−1^) (Figure 6B). On the other hand, MSC(M) and MSC(AT) presented a statistically significant increase in ALP activity compared to PDLSC after 21 days of culture under osteogenic conditions (MSC(M): [3.10 ± 0.26] × 10^−4^ μg·μL^−1^, MSC(AT): [2.70 ± 0.24] × 10^−4^ μg·μL^−1^) (Figure 6B). Despite this, the ALP activity of MSC(M) and MSC(AT) was not statistically significantly different, suggesting a similar osteogenic potential between these cells.

Gene expression levels of osteogenic/periodontal markers were evaluated by quantitative reverse transcription-polymerase chain reaction (qRT-PCR), namely runt-related transcription factor 2 (*Runx2*), collagen type I (*Col1A1*), *ALP*, *OPN*, *OC*, cementum protein-1 (*CEMP-1*) and periostin (*POSTN*) (Figure 7). After 21 days under osteogenic differentiation conditions, cells isolated from the different tissues upregulated the expression of osteogenic gene markers compared to the control (undifferentiated cells at day 0), confirming the successful osteogenic differentiation of MSC. Statistically significant differences in the expression levels of *Col1A1* (*p* < 0.001), *Runx2* (*p* < 0.01), *OC* (*p* < 0.001), *CEMP-1* (*p* < 0.05) and *POSTN* (*p* < 0.001) were observed between PDLSC and the bone-marrow and adipose tissue-derived cells (Figure 7). *OPN* (*p* < 0.001) and *ALP* (*p* < 0.001) gene expression of PDLSC was statistically enhanced compared to MSC(AT), however similar to MSC(M). Moreover, under the conditions tested, MSC(AT) presented the lowest osteogenic/periodontal potential, as suggested by the lower levels of *OPN*, *OC* and *POSTN* gene expression (Figure 7). Lastly, regarding *Col1A1*, *Runx2* and *CEMP-1* expression levels, MSC(AT) and MSC(M) did not present significant differences.

After 21 days of culture under osteogenic conditions, immunocytochemistry analysis of PDLSC, MSC(M), and MSC(AT) was performed (Figure 8). Results confirmed the expression of the common ECM proteins, laminin, and fibronectin, in all samples. Additionally, MSC derived from all sources stained positive for osteogenic and periodontal-related markers, namely OC, OPN and asporin. Interestingly, CEMP-1 was exclusively expressed by PDLSC (Figure 8).

Overall, the results presented in this work demonstrated that MSC isolated from PDL, bone marrow and adipose tissue were able to differentiate into the osteogenic lineage. However, under the conditions tested, significant differences between PDLSC, MSC(M), and MSC(AT) were observed concerning the levels of mineralization and osteogenic/periodontal gene expression, suggesting that PDLSC present a higher osteogenic potential.

## 4. Discussion

The results presented in our study suggest that PDL can be a promising cell source for periodontal tissue engineering applications. MSC derived from PDL (PDLSC) exhibited enhanced proliferative capacity and superior osteogenic/periodontal potential when compared to other adult MSC sources commonly used in regenerative medicine applications, namely MSC(M) and MSC(AT).

Successful periodontal regeneration requires coordinated regeneration of soft (PDL) and hard (cementum and alveolar bone) tissues. Limitations in current regeneration strategies remain due to outcome variability. Cell-based therapies have been investigated to improve clinical outcomes of periodontal regeneration [21]. In fact, several studies have supported the potential application of ex vivo expanded MSC for periodontal tissue regeneration [22]. For instance, MSC(M) injected into periodontal defects have demonstrated anti-inflammatory and immunomodulatory effects leading to tissue regeneration [9]. Although MSC(M) are considered the gold standard of cells for bone tissue engineering applications [22], dental-derived MSC, such as PDLSC, represent an attractive alternative cell source for periodontal regeneration due to their relative ease and less invasive access in comparison to MSC(M). Despite this, only few clinical trials are using PDLSC to treat periodontal intrabody defects [21]. Moreover, there is a lack of evidence to support the use of PDLSC for periodontal tissue engineering purposes compared to MSC derived from other adult tissues, such as MSC(M) and MSC(AT). Therefore, in this work, we evaluated the potential of PDLSC application in periodontal regeneration by comparing the osteogenic/periodontal potential of MSC derived from other adult tissues, including bone marrow, adipose tissue and PDL. Previous works have already demonstrated that PDLSC had a greater osteogenic potential than other dental derived-MSC, such as dental pulp stem/progenitor cells and stem/progenitor cells from human exfoliated deciduous teeth [23]. However, this is the first study performing a comprehensive characterization on the osteogenic potential of PDLSC and MSC isolated from the most common adult tissues exploited in clinical studies (MSC(M) and MSC(AT)).

To assess differences among MSC derived from different adult tissues, cell surface marker expression profiles were analyzed for cells at passages 3, 5, and 7 and compared by flow cytometry. As expected, and in line with previous reports, PDLSC [24,25], MSC(M) [26,27], and MSC(AT) [26,27] presented a positive expression of MSC-related markers CD29, CD44, CD73, CD90, and CD105. CD44 is a cell surface marker associated with MSC [28]. In fact, MSC(M) and MSC(AT) have been reported to present similar levels of expression of CD44 in early passages (passage 3 or lower) [29,30]. Moreover, reports have shown that MSC(M) can maintain a strong positive expression of CD44 despite cell passages [31]. However, under the conditions tested, our results demonstrated that CD44 expression of MSC(M) and MSC(AT) was negatively affected by passaging, whereas PDLSC were consistently positive for CD44 in early and late passages. Additionally, our results showed that Stro-1, a known MSC marker [16], was presenting low expressing levels by MSC derived from PDL, bone marrow and adipose tissue (<11.3%). In fact, it is unclear whether Stro-1 expression correlates with multipotency, and it has been reported that Stro-1 is not universally expressed by MSC derived from different sources [32,33]. Concerning CD146 expression, a cell surface marker previously applied in the identification of PDLSC [13] and commonly used as a marker for endothelial cells [34], both MSC(M) and MSC(AT) expressed low levels of CD146 in all passages studied (<7%). Interestingly, PDLSC robustly expressed CD146 presenting expression values of (67.40 ± 0.91)% in passage 3 and (69.51 ± 1.63)% in passage 5. However, the expression of CD146 by PDLSC decreased with passaging, reaching expression values of (24.06 ± 1.23)% in passage 7. Indeed, previous reports have demonstrated a reduced expression of CD146 in MSC derived from dental pulp and exfoliated deciduous teeth when cultured under continuous passage conditions [35]. Moreover, CD146^+^ PDLSC have been reported to exhibit higher colony-forming efficiency and proliferation rate when compared to CD146^−^ PDLSC [34]. Thus, we hypothesize that CD146 might contribute to enhanced cell proliferation of PDLSC. These results are in line with previous reports, which suggested that CD146 expression might be used as a marker for PDLSC [13]

Immunocytochemistry analysis of PDLSC, MSC(M) and MSC(AT) confirmed the presence of commonly expressed ECM molecules, including laminin and fibronectin. Overall, comparative analysis of PDLSC, MSC(M) and MSC(AT) revealed phenotypic similarities regarding the expression of MSC-related markers. Despite this, differences in PDLSC expression were observed, such as positive immunofluorescence stainings of asporin (a protein associated with the PDL [36]), OPN and OC (bone ECM proteins [37]) which were not expressed by bone marrow- and adipose tissue-derived cells. In fact, the positive stainings of osteogenic markers OPN and OC suggested that PDLSC might have a higher osteogenic potential, even when not cultured under osteogenic differentiation conditions (i.e., expansion medium). Still, additional comparative studies are necessary to assess if phenotypic differences between PDLSC and MSC derived from other sources have an impact on what concerns cell function and therapeutic potential.

Concerning the growth curves of PDLSC, MSC(M) and MSC(AT), substantial differences in terms of proliferative potential were noticed. Despite MSC(M) and MSC(AT) exhibiting similar growth patterns, cells from adipose tissue reached higher cell numbers, which is in line with previous research [27,38]. PDLSC exhibited significantly higher proliferation capacity reaching higher cell numbers compared to bone marrow- and adipose tissue-derived cells. Additionally, PDLSC also presented higher cell population doublings, regardless of the initial cell seeding density. In line with these findings, morphology assays revealed higher confluency in PDLSC culture after seven days in expansion conditions. Interestingly, immunocytochemistry analysis revealed that only PDLSC expressed OPN. In fact, previous studies demonstrated that OPN has chemoattractant properties that induce migration of MSC [39,40], neural stem cells [41], endothelial cells [42], and hematopoietic stem cells [43]. Furthermore, OPN promotes cell adhesion by interacting with several cell surface integrins. In this context, we hypothesize that positive OPN expression of PDLSC might be associated with the higher proliferation rate presented by PDLSC compared to MSC(M) and MSC(AT). In fact, this high proliferative capacity presented by PDLSC can be considered a technical advantage since MSC-based therapies depend on ex vivo expansion prior to in vivo administration in order to reach clinically meaningful cell numbers.

As expected, multipotency of PDLSC, MSC(M) and MSC(AT) was confirmed with positive stainings for multilineage differentiation across mesodermal lineages (osteogenic, adipogenic, and chondrogenic). The qualitative nature of in vitro stainings did not allow an accurate assessment of disparities in what concerns differentiation potential. Thus, further studies should consider staining’s quantification. However, it is relevant to note that PDLSC produced lower amounts of red-stained lipid vacuoles after adipogenic differentiation compared with MSC(M) and MSC(AT), suggesting a decreased adipogenic potential for cells obtained from PDL. Nevertheless, the Oil red O staining results did not allow a clear comparison between the three different cell types. Further quantitative assays are necessary to fully assess the adipogenic differentiation capacity of PDLSC, such as adipogenic gene expression analysis.

In what concerns osteogenic potential, PDLSC displayed increased osteogenic capacity in vitro compared to the other sources. We observed that, although MSC(M) and MSC(AT) presented similar values of calcium deposits after 21 days under osteogenic differentiation conditions, PDLSC significantly outperformed their amount of calcium. In fact, reports have shown that PDLSC have a phenotype characteristic of osteoblast-like cells [44,45], upregulating osteogenic marker genes and generating new bone following tooth extraction [46]. Surprisingly, ALP staining and its quantification demonstrated that the ALP activity of PDLSC did not increase significantly after culturing under osteogenic induction conditions. Yu and colleagues have shown that PDLSC are composed of a heterogeneous population of cells, presenting ALP^+^ and ALP^−^ cells [47]. They found that both ALP^+^ and ALP^−^ cells showed similar osteogenic potential with no observable difference in the amount of mineral deposits after osteogenic differentiation [47]. Therefore, the lack of ALP activity observed in this work might be due to the presence of a population of ALP^−^ cells. It is important to highlight that the lower levels of ALP activity of PDLSC did not compromise the osteogenic differentiation of these cells, as observed by the enhancement of mineralization and upregulation of osteogenic marker genes. Interestingly, the low levels of ALP activity in PDLSC might be related to the decreased adipogenic potential presented by PDLSC. In fact, ALP, besides being traditionally used as a marker of early osteogenesis, is involved in the control of intracellular lipid accumulation in adipocyte maturation. Thus, the absence of ALP may prevent formation of lipids in cells [48,49]. However, isolation of PDLSC from additional donors is required to understand the low levels of ALP activity observed by PDLSC.

Additionally, qRT-PCR results confirmed effective osteogenic differentiation of MSC after 21 days of culture under osteogenic differentiation conditions, independently of the tissue of origin. After osteogenic induction, MSC(AT) presented lower relative expression of osteogenic genes and downregulated *POSTN* gene expression compared to MSC(M) and PDLSC. Overall, these results suggested that, under the conditions of our study, adipose tissue-derived MSC possessed lower osteogenic capacity compared with PDLSC and MSC(M). In fact, previous comparative studies reported that MSC(AT) presented limited osteogenic potential compared to MSC(M), showing lower ALP activity, calcium content and expression of early and late osteogenic genes [26,27]. Although PDLSC upregulated the *ALP* gene after 21 days of osteogenic differentiation, enhancement of ALP activity was not observed. Despite the mRNA levels and protein activity of ALP tending to be correlated, there is not a linear correlation due to mRNA and protein regulation. The *Runx2* gene regulates downstream genes that determine the osteogenic phenotype and controls the expression of osteogenic marker genes such as *Col1A1*, *ALP*, *OPN*, and *OC* [50,51]. OC and OPN are non-collagenous proteins that play key roles in the biological and mechanical functions of bone [52,53]. Our results revealed that PDLSC presented higher upregulation of the *Runx2* gene than bone marrow and adipose tissue-derived cells. Furthermore, both MSC(M) and PDLSC exhibited higher upregulation of late osteogenic marker genes, *OC* and *OPN*. Collagen I is the most abundant protein in bone ECM (90% of the organic bone ECM) and is pivotal for matrix mineralization [52,53]. In fact, previous studies showed that, during osteogenic differentiation, PDLSC displays better collagen-forming capacity than MSC(M) [54]. After 21 days under osteogenic differentiation conditions, analysis of gene expression levels revealed higher upregulation of *Col1A1* and *CEMP-1* genes by PDLSC compared with MSC(M) and MSC(AT). Indeed, CEMP-1 has been identified as a novel cementum-specific protein and is strongly expressed by cementoblasts and their progenitors [55], including cells located near the blood vessels in the PDL [56]. Combined with the expression of CEMP-1 observed by immunocytochemistry analysis, enhanced upregulation of *CEMP-1* gene expression suggested that PDLSC may comprise a subpopulation of cementum progenitor cells, as previously proposed by McCulloch and Melcher [57] and further supported by recent studies [58]. Periostin is a matricellular protein with a fundamental role in bone and tooth tissue development and repair, namely remodeling of the collagen matrix [59] and maintenance of the integrity of the PDL in response to mechanical stresses [60]. Interestingly, after 21 days under osteogenic differentiation conditions, PDLSC presented higher *POSTN* gene expression levels compared with MSC(M) and MSC(AT), as assessed by qRT-PCR analysis. Immunofluorescence results corroborated the successful osteogenic differentiation of the three cell types, regardless of their tissue of origin. As expected, common ECM proteins, including laminin and fibronectin, were positively stained in all samples. Additionally, positive stainings for collagen I, OPN and OC were observed in MSC from all tissue sources. Concerning CEMP-1, only differentiated PDLSC displayed positive staining.

Overall, our comparative study demonstrates that PDLSC show an enhanced osteogenic/periodontal potential compared to bone marrow and adipose tissue-derived cells, showing promising results for periodontal tissue engineering applications [61,62,63]. Future studies should include more donors from each cell source to determine the impact of donor variability on cells’ osteogenic/periodontal potential. As a limitation of this study, cells isolated from the different tissues were not collected from the same donors as different procedures required to harvest PDLSC, MSC(M) and MSC(AT) are extremely invasive and are often carried out in different medical facilities. Moreover, future studies should include transcriptome analysis of PDLSC compared to MSC from other sources.

## 5. Conclusions

In summary, we have successfully isolated and characterized PDLSC, and we have shown that these cells exhibited enhanced proliferative and osteogenic/periodontal potential compared to MSC from other sources (bone marrow and adipose tissue). Our results showed that only PDLSC expressed OPN, a non-collagenous bone ECM protein important for cell proliferation and migration. Moreover, gene expression analysis revealed that PDLSC presented a significant enhancement in osteogenic/periodontal marker gene expression levels, such as *Runx2*, *OC*, *CEMP-1*, and *POSTN*, compared to MSC(M) and MSC(AT). Therefore, this work suggests that PDLSC are promising candidates for periodontal regeneration therapies, providing enhanced proliferative and osteogenic capacity. Future in vivo studies will be needed to assess if PDLSC retain their osteogenic and periodontal potential after administration and to determine the number of PDLSC required to obtain a clinical benefit.

## Data Availability

The authors declare that the data generated in the current study are available within the article or from the corresponding author upon reasonable request.

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
