# Peer review of "Enhanced Proliferative and Osteogenic Potential of Periodontal Ligament Stromal Cells"

_biomedicines, 2023, doi:10.3390/biomedicines11051352_

Round 1

Reviewer 1 Report

In the article entitled “Enhanced Proliferative and Osteogenic Potential of Periodontal Ligament Stromal Cells” by Laura Alves and colleagues, the authors describe 3 MSC line derived from different sources (bone marrow, adipose tissue and PDL). The main idea is simple and clear. In this study, they compared the morphology, immunophenotype, proliferation rate and trilineage differentiation potential. This work is not very big, but may have significance in practice.  

However I have some questions and comments.

1.      Why did the authors not compare with MSCs from dental pulp?

2.      Two donors from each cell type (PDLSC, MSC(M) and MSC(AT)) were used in this work. It is not so well for result verification.

3.      Why analyze the cytoskeleton and metabolic activity?

4.      Table 1. Col I = Col1A1?

5.      It is incorrect to estimate the percentage of positive cells in this case. We see on the histograms a single population of cells (only one peak in Supplementary). We don't see two separate different populations. The authors could estimate the MFI, but not the percentage. The percentage of cells can only be estimated if there are two separate populations on the histogram: positive and negative.

6.      Line 477-478. The bound dye can be extracted and evaluated on a spectrophotometer.

7.      Number of experiments (n, independent experiments) is unknown on figures 1,2,3,5,8.

Reviewer 2 Report

The manuscript represents a comparative study of three cell types of mesenchymal stromal origin for regeneration of periodontium tissue. The experimental work performed by the authors is of reasonable quality. However, the content restructuring is advised to improve comprehensiveness of the manuscript. Additionally, the title of the manuscript does not reflect the experimental work described. Most importantly, the number of donors/biological trials needs to be clarified here to determine the relevance of statistical data processing and overall conclusions. The manuscript then needs to be adjusted based on the offered clarification. The detailed comments are provided below:

Title:

From the experimental work it seems that the data were compared at one single passage (P3) from one (at most two) donors for 3 technical replicates (3 cell seedings from that passage). The number of donors/biological trials presented here does not permit statistical treatment of data and cannot lead to any firm conclusions. The title needs to specify that this is an observational study, where 1 donor (for some experiments 2 donors) was used to make comparisons. Ideally, 3 different cell donors with the cells in the same passage should be used. Alternatively, 3 different cell passages from one donor can also suffice (given the heterogeneity between the passages is not very high). The title needs to reflect these limitations!!!

Abstract:

First sentence is ambiguous. The Authors need to specify here regeneration of what tissues they referring to (periodontium). The statement cannot be applied to other tissues.

Introduction:

This section is written very well, clearly defining the challenge and offered solutions.

Materials and Methods:

Please include a figure with a time-line including which experiments were performed on which day. This will give a nice overview to the reader.

Section 5.2.: Are the same donors used for bone marrow and adipose tissue-derived MSCs? What were the genders of the donors?

Section 5.3.1.: Ascorbic acid is not a stable chemical. Which salt of the AA was used?

l.138: “… stained with alkaline phosphatase (ALP)”. Did the Authors mean stained for ALP?

Please combine all the osteogenic differentiation related assays (5.8.2. and 5.8.3.) as subsections of section 5.3.1. Otherwise, all the techniques are scattered.

5.9. Statistical analysis: please clarify what statement “Each experiment was conducted in triplicate” means. How many donors, how many passages from each donor (biological trials) and how many cell seedings (technical replicates) from each biological trial were used. This info is crucial, since it implies the strength of the presented data.

Results:

l. 279-283: When the Authors refer to Figure 3 and state that Alizarin red, oil red o staining were increased/reduced/the same, what are these conclusions based on? Was there quantification of the staining performed or it is purely observational? This needs to be stated and described as limitation.

Figure 1 legend: since only 2 donors are used for each passage the statistical treatment is not relevant here. 

Please put in each figure legend: number of donors, number of biological trials (summarized results of different passages), number of technical replicates (seedings per each passage/donor) !!! If the data is not summary of 3 different donors/ 3 different passages, remove statistical data treatment, since it has no relevance here.

Figure 2.  Please put in figure legend on which day the ECM proteins were stained.

Please put in all figure legends time point of the analysis !!!

Please combine/move findings in Figure 6 with the differentiation results, otherwise it is out of context there.

As a recommendation, to make the results easy to follow, it makes more sense to start with immunophenotype of cells (Fig.1), then go to morphological features (Fig.5) and ECM proteins in expansion medium (Fig. 2), then proceed with proliferation data (Fig.4) and as a final step - differentiation data (Fig.3 + Fig.6) and qRT-PCR (Fig.7), at the end ECM proteins in differentiation medium (Fig.8). Otherwise the data is scattered and not structured well.

Throughout the whole manuscript remove sentences like “significant differences” for data, where no statistical data treatment could be performed.

Discussion:

Is very well written, however, needs to be modified, given that the data cannot be backed up by statistical treatment. Please also specify what limitations the study has.

Reviewer 3 Report

The authors aime to characterize the osteogenic/periodontal potential of periodontal ligament derived stromal cells (PDLSC) in comparison to MSC’s derived from the bone marrow (MSC(M)) and adipose tissue (MSC(AT)) and state that regeneration of a functional periodontium has never been achieved, which emphasizes on the relevance of their work. The introduction misses some relevant references on the subject. The materials and methods are as expected. The manuscript is in general well done and well presented, but there are several issues that require clarification before being able to accept the manuscript:

1.     On figure 1, the bar images have to be replaced by the histograms on the supplementary images plotted all together for comparative purposes. From the bar-images, it is necessary to retain only the ones with significative differences

2.     On figure 2, please describe the reasons for the different nuclei sizes on PDLSC MSC(M) and MSC(AT). In addition, it would be appreciated if the authors also describe the reasons for the size differences observed in the MSC(M) panels where nuclei stained with DAPI are visible at 50x magnification

3.     The data in figure 6 shows a discrepancy between the calcium concentration and the ALP activity? In fact, on PBLSC there is an ~8-fold increase in Calcium upon differentiation, with no change on ALP activity, whereas in MSC(M) and MSC(AT) there is an increase in Calcium concentration although somehow more modest for MSC(AT) and reaching calcium concentrations significantly lower that for PDLSC’s, with increases over 3-fold in ALP activity. Can these differences be due to a kinetic problem, in particular since the MSC(M) and MCS(AT) proliferate at a much lower rate than the PCLSC’s, as demonstrated on figure 4. Or is it conceivable that the differentiation kinetics is distinct for the cells from these three sources, due in part to their different proliferation ratios? It would be nice the author’s possible explanations to this issue: These differences might simply be a kinetics difference and not a higher capacity of PDLSC to generate bone

4.     Regarding Figure 7: please explain the discrepancy between ALP mRNA and activity on PDLSCs, and between MSC(M) and MSC(AT) there the activity is similar but the amount of RNA is > than 3-fold different

5.     Regarding Figure 7: please comment on the OPN and OC results on the PCR and the staining’s on figure 2 of the day 0 cells

6.     On Figure 8, the quality of Col I, CEMP-1, ASPN, OPN and to some extent OC on the PDLSC cells it should be higher. In MSC(M) there are large differences on the size of the nuclei on plates with 50 µm scale bar (see a similar comment for figure 2)

Small issues:

7.     On line 251 states “CD80 was negatively expressed” the statement is simply not correct. The expression levels of CD80 can be high, low, or expression of CD80 has been reduced/silenced, etc. please change

8.     On lines 260-262 it is stated “only PDLSC expressed Stro-1 (a known MSC marker), asporin, osteocalcin (OC) and osteopontin (OPN). MSC(M) and 261 MSC(AT) did not express Stro-1, asporin, OC and OPN (Figure 2)“. Basically, I assume you mean that expression of the known MSC marker Stro-1, asporin, osteocalcin (OC) and osteopontin (OPN) can be detected on undifferentiated cells from PDLSC origin, but not in MSC(M) or MSC(AT)-derived undifferentiated cells. 

9.     On lines 277-278 it is stated: “however PDLSC presented enhanced mineralization, observed by VK and Alizarin Red stainings“. Has the mineralization been quantified on the images of figure 3?, otherwise the text should be changed accordingly.

10.  Figure 4 is too complex with redundant data shown in panels A and B. I would suggest to take the graph on 4B and split in into two with the same scale, one of them (the new 4A that would contain the 1500 cells/cm^3 and the other corresponding now to 4B that would contain the 3000 cells/cm^3. Since the population doublings (4C) correspond directly to the number of cells, the significance analyses could be directly added to the new 4A and 4B, otherwise if the authors prefer, they could keep it as the actual 4C.

11.  On line 418 where it says instances it should say instance

12.  On line 446: was negatively expressed (see comment 7) 

Round 2

Reviewer 2 Report

The manuscript represents a comparative study of three cell types of mesenchymal stromal origin for regeneration of periodontium tissue. The experimental work performed by the authors is of reasonable quality. However, the content restructuring is advised to improve comprehensiveness of the manuscript. Additionally, the title of the manuscript does not reflect the experimental work described. Most importantly, the number of donors/biological trials needs to be clarified here to determine the relevance of statistical data processing and overall conclusions. The manuscript then needs to be adjusted based on the offered clarification. The detailed comments are provided below:

Title:

From the experimental work it seems that the data were compared at one single passage (P3) from one (at most two) donors for 3 technical replicates (3 cell seedings from that passage). The number of donors/biological trials presented here does not permit statistical treatment of data and cannot lead to any firm conclusions. The title needs to specify that this is an observational study, where 1 donor (for some experiments 2 donors) was used to make comparisons. Ideally, 3 different cell donors with the cells in the same passage should be used. Alternatively, 3 different cell passages from one donor can also suffice (given the heterogeneity between the passages is not very high). The title needs to reflect these limitations!!!

Author response: We acknowledge the reviewer for making this comment. We did not perform only one single passage from one donor with three technical replicates. In fact, for each cell source, we have used two different donors and performed three independent experiments for each donor with three technical replicates each. Since we only had 6 different donors (2 donors for each cell source), we tried to improve the statistical power by performing three different independent experiments for each donor (so total of 6 independent experiments for each cell source). Moreover, each experiment was performed with three technical replicates. Therefore, in the end, for each time-point for each cell donor, there would be 9 data points (3 experiments x 3 technical replicates). Nevertheless, we also understand that future studies must include more overarching biological replicates to evaluate the impact of donor variability on cells’ osteogenic potential. We have acknowledged this limitation and expect that this has been adequately highlighted in the “Materials and Methods” and “Discussion” sections (Page, 5, line 223-225; Page 19, line 613-615).

Referee: Usually “three independent experiments” are overcome via either a minimum of 3 donors or using 3 independent cell passages (e.g., P3, P4 and P5). Therefore, it still remains unclear, how the authors used, for example, Passage 3 from 2 donors as 3 independent experiments. In my understanding, the cells are isolated from a donor and then expanded/passaged. Therefore Passage 3 can be repeated once from each donor. Unless the authors found a way to overcome this challenge. Otherwise, the statistical treatment for experiments repeated for one passage from 2 donors makes no sense to me. Please clarify. This applies to the title too.

Abstract:

First sentence is ambiguous. The Authors need to specify here regeneration of what tissues they referring to (periodontium). The statement cannot be applied to other tissues.

Author response: We acknowledge the reviewer for making this comment. The revised version of the manuscript specifies that the periodontal tissue is the object of regeneration so that the first sentence reflects the aim of this work (Page 1, line 15-16).

Referee: Accepted

Introduction:

This section is written very well, clearly defining the challenge and offered solutions.

Author response: We thank the reviewer for the thorough and attentive reading of the  manuscript.

Materials and Methods:

Please include a figure with a time-line including which experiments were performed on which day. This will give a nice overview to the reader.

Author response: We thank the reviewer for this suggestion. We have included a figure with a time-line including all the experiments performed, as suggested by the reviewer (Scheme 1).

Author Response

Author response: We thank the reviewer for this comment. In this work, cells were isolated from two donors, cryopreserved, thawed and then expanded. For each independent experiment, a cell vial was thawed and expanded to reach a passage between 3 and 5. We have corrected this information in the manuscript, clarifying that cells between passage 3 and 5 (P3, P4 and P5) were used for each donor and cell source (Page 5, line 223-225). For each cell source, we have used two different donors and for each donor, we have performed three independent experiments with different cell vials thawed and expanded to also different cell passages (P3-P5), with three technical replicates each.

Reviewer 3 Report

ok

Author Response

We thank the reviewer for the careful reading of the manuscript. 

Round 3

Reviewer 2 Report

In Scheme 1: Please change "Cell Membrane" to "Cell F-actin"

Phalloidin does not stain cell membrane! and please put a picture of stained actin fibers!

Author Response

We thank the reviewer for the comments provided. We have revised the manuscript as suggested by the reviewer.